# Efficient and Accurate Top-$K$ Recovery from Choice Data

**Duc Nguyen**[1]

[1]Department of Computer and Information Science, University of Pennsylvania.

## Abstract

The intersection of learning to rank and choice modeling is an active area of research with applications in e-commerce, information retrieval and the social sciences. In some applications such as recommendation systems, the statistician is primarily interested in recovering the set of the top ranked items from a large pool of items as efficiently as possible using passively collected *discrete choice data*, i.e., the user picks one item from a set of multiple items. Motivated by this practical consideration, we propose *the choice-based Borda count algorithm* as a fast and accurate ranking algorithm for *top $K$-recovery* i.e., correctly identifying all of the top $K$ items. We show that the choice-based Borda count algorithm has optimal sample complexity for top-$K$ recovery under a broad class of *random utility models*. We prove that in the limit, the choice-based Borda count algorithm produces the same top-$K$ estimate as the commonly used Maximum Likelihood Estimate method but the former's speed and simplicity brings considerable advantages in practice. Experiments on both synthetic and real datasets show that the counting algorithm is competitive with commonly used ranking algorithms in terms of accuracy while being several orders of magnitude faster.

## 1 INTRODUCTION

The research on discrete choice modeling and learning to rank has received a lot of interest in recent years thanks to the growing availability of discrete choice data generated by e-commerce platforms, search engines and the social sciences. In the discrete choice setting, when presented with a set of items, also referred to as *menu*, the user picks the most preferred item. Discrete choice data is an intermediate between pairwise comparison data and full ranking data. In many settings such as e-commerce and political surveys, a large quantity of passively collected data is in the form of discrete choice data, e.g., consumers choosing to buy a product when presented with a catalogue of items, voters picking a favorite candidate from a pool of candidates.

In this paper, we focus on the problem of learning to rank using choice data. Specifically, we are interested in the top-$K$ recovery problem, i.e., identifying the set of the top $K$ items out of a universe of $n$ items, using *passively collected choice data*. This problem has many useful applications. For example, in e-commerce applications, marketers are interested in finding the set of the best items based on how consumers make purchasing decisions. In the social sciences, political scientists are interested in determining the most preferred candidates among a pool of candidates using survey questionnaires.

To ground our theoretical discussions, we posit that the choice data is generated according to a probabilistic choice model- when presented with a menu of items $S$, the user makes a non-deterministic decision, picking a single item $i$ from $S$ with some probability $p_{i|S}$. More specifically, we assume our choice model falls within the class of Random Utility Models with Independently and Identically Distributed noise (IID-RUMs), described in detail in section (2). IID-RUMs are an expressive and flexible framework that can be used to model pairwise comparison data, discrete choice data as well as full ranking data. For example, the Multinomial Logit (MNL) model is one of the most commonly used IID-RUMs to model discrete choice data [Train, 2009].

**Our motivation:** While expressive, random utility models also pose hard computational problems. For example, many models within the class of IID-RUMs with the few exceptions such as the MNL model do not admit analytical expression for the choice probabilities (while the pairwise comparison probabilities can be evaluated easily), limiting inference to MCMC-based algorithms. However, sampling-based algorithms can be time inefficient when running on

*Accepted for the 38th Conference on Uncertainty in Artificial Intelligence* (UAI 2022).

large choice datasets with many items and menus. Furthermore, most classical inference algorithms assume a parametric model generating the choice data. In practice, it is often hard to verify if the data comes from a specific parametric model. Therefore, developing efficient ranking algorithms that are robust to model misspecification is of timely interest.

Motivated by these considerations, we study the generalization of a simple yet powerful counting algorithm for ranking- Borda count - to the discrete choice setting. The Borda count algorithm itself has a long history, dating back to the 18th century and its analysis has been instantiated in various contexts such ranking from pairwise comparisons in Rajkumar and Agarwal [2014], Shah and Wainwright [2017]. Our work, however, is the first to study the theoretical guarantees of Borda count in the *discrete choice setting* under a broad class of discrete choice models.

**Our contributions:**

- In Section 4 and Section 5, we show that the choice-based Borda count algorithm needs $\theta(n \log n)$ samples in order to exactly recover all of the top $K$ items using choice data. We further show that this sample complexity is optimal for a broad class of IID-RUMs. This hinges on a fundamental property shared by many IID-RUMs which we term *Borda consistency*.

- In Section 6, we study the effect of the menu size $m$ on the sample complexity for top $K$ recovery. For the special case of the MNL model, which is a common assumption in the ranking literature, we present an asymptotic characterization of the optimal sample complexity for top $K$ recovery in terms of $m$. This bound monotonically decreases, but at a decreasing rate, with $m$. This suggests that there is a benefit to increasing the menu size but such benefit comes with diminishing returns. To the best of our knowledge, this result is the first of its kind in the choice modeling and ranking literature.

- In Section 7, we study the connections between the choice-based Borda count algorithm and two commonly used top-$K$ recovery algorithms: Maximum Likelihood Estimate under MNL assumption (MNL-MLE) and Spectral Ranking [Negahban et al., 2017, Maystre and Grossglauser, 2015, Agarwal et al., 2018]. We prove that the choice-based Borda count algorithm and MNL-MLE produce the same top-$K$ estimate in the limit of infinite data, even if the data has not been generated by an IID-RUM. On the other hand, Spectral Ranking does not in general give the same estimate as the choice-based Borda count algorithm/MNL-MLE even with infinite data.

- In Section 8, We show through empirical experiments that the choice-based Borda count algorithm is competitive in terms of accuracy with both MNL-MLE and

Spectral Ranking while being several orders of magnitude faster. This highlights the advantage of the choice-based Borda count algorithm in applications where the statistician is primarily interested in efficiently and accurately identifying the top items.

## 1.1 RELATED WORKS

Our work falls within the literature on learning to rank under Random Utility Models (RUMs). There has been a substantial amount of work on learning to rank under Random Utility Models and mixtures of Random Utility Models using *full ranking data* [Parkes et al., 2012, Azari Soufiani et al., 2013b,a, Soufiani et al., 2014, Zhao et al., 2016, Zhao and Xia, 2019]. Furthermore, most classical ranking methods assume that the data is generated by a well specified RUM. To the best of our knowledge, our paper is the first to propose a method for top-$K$ ranking under a broad class of RUMs using *passively collected choice data alone*.

The related literature on ranking from pairwise comparisons is vast and we can only refer the interested reader to adjacent problems such as *active top-$K$ recovery* from pairwise comparisons [Busa-Fekete et al., 2013, Agarwal et al., 2017, Mohajer et al., 2017, Falahatgar et al., 2017, 2018, Heckel et al., 2019]; top-$K$ recovery from pairwise comparisons [Chen and Suh, 2015, Shah and Wainwright, 2017, Chen et al., 2019]; top-$K$ recovery from *$m$-wise sorted data* (full rankings among some $m$ items) [Jang et al., 2017, Chen et al., 2020].

Closest to our work is the analysis of Borda count by Shah and Wainwright [2017] who showed that it is optimal for top-$K$ recovery from *pairwise comparisons*. Our work complements theirs by showing that the choice-based Borda count is optimal even in the *general choice setting*. To this end, we obtain in Section 4 sample complexity upper and lower bounds that are *both more general and refined* than those given by Shah and Wainwright [2017]. We also study in Section 6 the effect of the menu size on the sample complexity. To the best of our knowledge, our paper presents the first asymptotic characterization of the sample complexity for ranking from choice data in terms of the menu size under the very commonly used MNL model. Operating on *$m$-wise sorted data*, Jang et al. [2017] showed that the optimal sample complexity for top-$K$ recovery under the Plackett Luce model[1] scales with $O(\frac{1}{m})$. Our results complement theirs by showing that the sample complexity for top-$K$ recovery from *discrete choice data* scales as $O(1 + \frac{1}{m})$. Furthermore, the choice-based Borda count algorithm is different from the Spectral-MLE algorithm studied there which is specialized to the Plackett-Luce model.

---

[1]Within the ranking literature, Plackett-Luce (PL) is a class of distributions over permutations, induced by the IID-RUM with standard Gumbel noise.

## 2 NOTATIONS AND PROBLEM FORMULATION

Let there be $n$ items in the universe. Each item $i$ has a *deterministic and hidden* utility, also referred to as partworth, $U_i$ for $i = 1, \ldots, n$. Let us assume the non-degenerate case where no two items have identical partworths. Without loss of generality, we also assume that $U_{\max} = U_1 > U_2 > \ldots > U_n = U_{\min} > 0$. Let $\mathcal{S}_K^* = \{1, \ldots, K\}$ denote the set of $K$ items with the highest parthworths.

Items are presented to the consumer in a set $S$, also referred to as menu, of size at least 2. When $S$ is presented to the consumer, the perceived utility of each item $i \in S$ is the sum of its parthworth and a random noise term: $X_i = U_i + \epsilon_i$ where the $\epsilon_i$'s are independently and identically distributed according to an *unknown* universal noise distribution $D$. The consumer then picks the item $i$ with the highest perceived utility among all the items in $S$. Such a choice model is referred to as a random utility model with independent and identically distributed noise (IID-RUM). In short, a choice model $\rho$ within the class of IID-RUMs is parametrized by a set of partworths $\{U_1, \ldots U_n\}$ and noise distribution $D$.

As an overload of notation, we will also use $\rho(i|S)$ to denote the probability that a consumer picks item $i$ from menu $S$ under choice model $\rho$. By definition, $\rho(i|S) = \mathbb{P}(X_i > X_k \ \forall k \in S \backslash \{i\})$. For simplicity, we consider a fixed menu size $m$. However, our analysis can be easily extended to account for a mixture of menu sizes.

A choice sample is a (menu, item) tuple $(S, y)$ where the consumer chooses item $y$ from menu $S$. A choice dataset is a set of choice samples. A top-$K$ recovery algorithm takes in a choice dataset and returns an estimate of the top $K$ items, $\hat{\mathcal{S}}_K$. The goal is to exactly recover the top $K$ items and the performance metric of interest is the 0-1 loss: $L_{01}(\hat{\mathcal{S}}_K, \mathcal{S}_K^*) = \mathbf{1}[\hat{\mathcal{S}}_K = \mathcal{S}_K^*]$ [2].

We emphasize that as opposed to the *top-$K$ ranking* problem, the objective of the top-$K$ recovery prolem is to accurately identify the set of the top $K$ items, while allowing for mis-ranking among these items.

## 3 THE CHOICE-BASED BORDA COUNT ALGORITHM

As discussed previously, the general counting approach referred to as Borda count has a long history and has been instantiated in various contexts such as ranking from pairwise comparisons. Here, we instantiate the Borda count approach to the more general discrete choice setting. This is shown in Algorithm 1.

As would be expected, the algorithm essentially tallies the

---

[2] $\mathbf{1}$ is the indicator function and the equality is with respect to set equality.

---

number of observed 'wins' by each item and finally ranking the items by their number of wins, returning the top $K$ items. As with other versions of the Borda count approach, the algorithm is simple and easy to implement; and very efficient in practice. This makes the choice-based Borda count algorithm appropriate in settings where the statistician is primarily interested in efficiently and accurately recovering the set of the top items from a large pool of choice data.

---

**Algorithm 1** The choice-based Borda count algorithm

**Input:** Choice dataset $\mathcal{B} = \{(S_l, y_l)\}_{l=1}^N$
**Output:** Top-$K$ estimate $\hat{\mathcal{S}}_K$

1: For each item $i = 1, \ldots, n$
2:     Compute the number of times $i$ gets chosen:
3:     $\hat{W}_i := \sum_{l=1}^N \mathbf{1}[y_l = i]$
4: Return the set of $K$ items corresponding to the highest $\hat{W}_i$'s. Ties are broken arbitrarily.

---

## 4 SAMPLE COMPLEXITY BOUND

In this section, we present the sample complexity of the choice-based Borda count algorithm for top-$K$ recovery. We first formalize our sampling model in Section 4.1. In Section 4.2, we characterize the class of IID-RUMs under which the choice-based Borda count can successfully identify all of the top $K$ items via a theoretical quantity we term the *generalized Borda score*. The main theorems on the sample complexity of the choice-based Borda count algorithm are presented in Section 4.3.

### 4.1 THE SAMPLING MODEL

Let $\mathcal{C}^{(m)}$ be the set of *all menus* of size $m \geq 2$ (i.e., $|C^{(m)}| = \binom{n}{m}$). Additionally, let $\mathcal{C}_i^{(m)}$ be the set of all menus of size $m$ containing item $i$ (i.e., $|\mathcal{C}_i^{(m)}| = \binom{n-1}{m-1}$). We consider a multiple-round uniform sampling model with $R$ rounds of sampling in total. In each round $r = 1, \ldots, R$, each menu $S \in \mathcal{C}^{(m)}$ is independently offered with probability $p > 0$. Let $\hat{\mathcal{C}}^{(m,r)}$ denote the set of menus of size $m$ that are offered in round $r$. If offered menu $S$, the user responds with a random choice $y_S^{(r)}$, where

$$\mathbb{P}(y_S^{(r)} = i) = \rho(i|S).$$

It is easy to check that we have, in expectation, $pR\binom{n}{m}$ samples over $R$ rounds.

As a practical example, this sampling procedure can be used to design online political surveys. Suppose that there are $R$ voters willing to take part in answering survey questionnaires to determine support for $n$ political candidates. Fix a ballot size $m$. For each voter, each ballot of size $m$ is

independently presented to that user with probability $p$. For each ballot, the voter picks one favourite candidate.

## 4.2 THE GENERALIZED BORDA SCORE

For each item $i$, define the following theoretical quantity, which we term the *generalized Borda score*:

$$\tau_i^{(m)} = \frac{1}{\binom{n-1}{m-1}} \cdot \sum_{S \in \mathcal{C}_i^{(m)}} \rho(i|S) \,.$$

Intuitively, the generalized Borda score is the expected probability that an item $i$ is chosen from a menu $S$ where $S$ is uniformly sampled from $\mathcal{C}_i^{(m)}$. Note that $\tau_i^{(m)} \in [0, 1]$ for all $i \in [n]$. The generalized Borda score is interesting to us because *for a large class of IID-RUMs*, the order among the items with respect to the generalized Borda scores is the same as that with respect to the partworths. Therefore, it suffices to rank the items by their generalized Borda scores to recover the items with the highest partworths. Formally, we can characterize this class of IID-RUMs using a property we term *Borda consistency*.

**Definition 1.** *An IID-RUM $\rho$ satisfies Borda consistency if for any two items $i, j$ and any menu size $m \geq 2$,*

$$\tau_i^{(m)} > \tau_j^{(m)} \Leftrightarrow U_i > U_j \,.$$

The follow lemma establishes that many commonly used IID-RUMs such as the MNL (Gumbel distributed noise) and the Probit (Normal distributed noise) model satisfy Borda consistency.

**Lemma 4.1.** *All IID-RUMs whose noise distribution has absolutely continuous density function and support on the real line satisfy Borda consistency.*

In the supplementary materials, we will show that Borda consistency is satisfied by an even broader class of IID-RUMs that include other commonly used models such as the IID-RUM with exponentially distributed noise. Intuitively, this stems from the property enjoyed by many IID-RUMs: for any two items $i, j$ where $U_i > U_j$, $\rho(i|S) > \rho(j|S) \; \forall S \in \mathcal{C}^{(m)} : i, j \in S$; and $\rho(i|S \cup \{i\}) > \rho(j|S \cup \{j\}) \; \forall S \in \mathcal{C}^{(m-1)} : i, j \notin S$. To the best of our knowledge, this fundamental property that holds across a very broad class of IID-RUMs has not been previously decribed in the literature and may be useful to future works exploring the intersection of ranking and choice modeling.

## 4.3 EXACT TOP-$K$ RECOVERY

Having established Borda consistency as a property enjoyed by many IID-RUMs, we will now present the finite sample guarantees of the choice-based Borda count algorithm for top-$K$ recovery that holds for all choice models in this broad class of IID-RUMs. To show that the choice-based Borda count algorithm accurately identifies all of the top $K$ items with high probability, it suffices to bound the probability that the algorithm mistakenly ranks an item $j \notin \mathcal{S}_K^*$ higher than another item $i \in \mathcal{S}_K^*$. Specifically, we want to bound the following probabilities.

$$\mathbb{P}(\hat{W}_j > \hat{W}_i) \quad \forall i \in \mathcal{S}_K^*, j \notin \mathcal{S}_K^* \,,$$

where $\hat{W}_i$ is defined in Algorithm (1). Considering this, the fundamental hardness of top-$K$ ranking lies in distinguishing between the $K$-th and $K + 1$-th best item, and therefore depends on the gap between their generalized Borda scores:

$$\Delta_K^{(m)} = \tau_K^{(m)} - \tau_{K+1}^{(m)} \,.$$

The smaller this gap, the more data the algorithm requires in order to correctly separate between the top $K$ and the bottom $n - K$ items. Building on this intuition and generalizing to any pair of items $(i, j)$ where $\tau_i > \tau_j$, we obtain the following upper bound on $\mathbb{P}(\hat{W}_j > \hat{W}_i)$.

**Lemma 4.2.** *Consider an IID-RUM that satisfies Borda consistency per Definition 1. Assume input choice data with menu size $m$ is generated according to the sampling model described in Section 4.1. For any two items $i$ and $j$ where $\tau_i^{(m)} > \tau_j^{(m)}$, the choice-based Borda count algorithm satisfies*

$$\mathbb{P}(\hat{W}_j > \hat{W}_i) \leq \exp\left( \frac{-3pR\binom{n}{m}m(\tau_i^{(m)} - \tau_j^{(m)})^2}{8n(\tau_i^{(m)} + \tau_j^{(m)})} \right) \,.$$

The proof of Lemma 4.2 uses a standard concentration inequality argument based on Bernstein's inequality (cf. Theorem 2.8.4 Vershynin [2018]). The lemma itself states that, for each pair $i \in \mathcal{S}_K^*, j \notin \mathcal{S}_K^*$, if $pR\binom{n}{m} \geq \frac{8n \log n(\tau_i^{(m)} + \tau_j^{(m)})}{m(\tau_i^{(m)} - \tau_j^{(m)})^2}$, then $\mathbb{P}(\hat{W}_j > \hat{W}_i) = O(\frac{1}{n^3})$. We also have the following lemma which presents an upper bound on the item-dependent term $\frac{\tau_i^{(m)} + \tau_j^{(m)}}{(\tau_i^{(m)} - \tau_j^{(m)})^2}$.

**Lemma 4.3.** *Consider an IID-RUM that satisfies Borda consistency per Definition 1. For any $K$, we have*

$$\frac{\tau_K^{(m)} + \tau_{K+1}^{(m)}}{\Delta_K^{(m)2}} = \max_{i \in \mathcal{S}_K^*, j \notin \mathcal{S}_K^*} \left\{ \frac{\tau_i^{(m)} + \tau_j^{(m)}}{(\tau_i^{(m)} - \tau_j^{(m)})^2} \right\} \,.$$

By combining the two lemmas above and applying union bound over all pairs $i \in \mathcal{S}_K^*, j \notin \mathcal{S}_K^*$, we obtain the following sample complexity bound for exact top-$K$ recovery:

**Theorem 4.4.** *Assume the conditions of lemma (4.2). Given sufficiently large $p, R$ such that $pR\binom{n}{m} \geq \frac{8n \log n}{m\Delta_K^{(m)2}} \cdot (\Delta_K^{(m)} +$*

$2\tau_{K+1}^{(m)}$), *the choice-based Borda count algorithm correctly identifies all of the top $K$ items with probability at least $1 - O(\frac{K^2}{n^2})$.*

The reader may also recognize that $\Delta_K^{(m)} + 2\tau_{K+1}^{(m)}$ is simply $\tau_K^{(m)} + \tau_{K+1}^{(m)}$. The former presentation is, however, useful in highlighting the main quantities that will also reappear in our matching lower bound. In summary, the choice-based Borda count algorithm has the following sample complexity for exact top-$K$ recovery:

$$O\left( \frac{n \log n}{m \Delta_K^{(m)}} \cdot (1 + \frac{\tau_{K+1}^{(m)}}{\Delta_K^{(m)}}) \right).$$

This shows that overall, we only need $O(n \log n)$ examples to recover the top $K$ items from choice data with high accuracy. Our upper bound (and matching lower bound to be shown) can be seen as both *generalization and refinement* of Theorem 1 of Shah and Wainwright [2017]. Under the pairwise comparison setting ($m = 2$), we can simply upper bound $\tau_{K+1}^{(m)} \le 1$ and recover the (optimal) sample complexity $O(\frac{n \log n}{\Delta_K^{(2)2}})$ of Borda count obtained by Shah and Wainwright [2017]. The analysis approach there, however, is insufficient to produce an optimal sample complexity bound in the discrete choice setting. Note also that there can be combinatorially many realizations of the data in the discrete choice setting as $|\mathcal{C}^{(m)}| = \binom{n}{m}$. Our proof therefore requires considerably more effort. Our bound also shows that the sample complexity depends not only on the gap $\Delta_K^{(m)}$ between the $K$-th and $K + 1$-th item, but also the relative 'strength' of the $K + 1$-th item, as captured by the $\frac{\tau_{K+1}^{(m)}}{\Delta_K^{(m)}}$ term.

In general, the factors $\tau_{K+1}^{(m)}$ and $\Delta_K^{(m)}$ don't admit closed form expressions because both are sums of $\binom{n-1}{m-1}$ terms. The reader may also recognize that these parameters also depend on the menu size $m$, the partworth parameters and the noise distribution. In the next section, we will show a *matching lower bound* in terms of the same parameters, establishing the optimality of the choice-based Borda count algorithm, and discuss why the exact relation between $\Delta_K^{(m)}$, $\tau_{K+1}^{(m)}$ and the model parameters remains elusive.

Often in practice, we may tolerate some error for top-$K$ ranking by allowing the algorithm to misidentify, up to a threshold, some number of items. This is known as *approximate top-$K$ recovery*. We include detailed discussions of this problem in the supplementary materials and show that the choice-based Borda count algorithm also has *optimal sample complexity* for approximate top-$K$ recovery under the broad class of IID-RUMs that satisfy Borda consistency.

# 5 INFORMATION-THEORETIC LOWER BOUND

In this section, we will show that the choice-based Borda count algorithm enjoys optimal sample complexity by furnishing a matching lower bound. To show a lower bound, we will construct a special subclass of the MNL family where any estimator requires $\Omega(n \log n)$ examples in order to exactly recover the top $K$ items. We defer detailed descriptions of this model to the supplementary materials while stating the main results as follows.

**Theorem 5.1.** *Consider the sampling model described in Section 4.1. There exists a class of MNL models such that for $n \ge 20$, if $pR\binom{n}{m} \le \frac{n \log n}{8} \cdot \frac{\tau_{K+1}^{(m)} + \Delta_K^{(m)}}{m \Delta_K^{(m)2}}$ then any estimator fails to correctly identify all of the top $K$ items with probability at least $\frac{1}{12}$.*

The proof of Theorem (5.1) first reduces the problem of exact top-$K$ recovery to a multiple hypothesis testing problem and then applies Fano's lemma [Cover, 1999]. Each hypothesis in the testing problem corresponds to an MNL model. Within each model, the set of the top $K$ items always includes items $1, \ldots, K - 1$. However, the index of remaining item in the top-$K$ set is different for each model (i.e., there are $n - K + 1$ different models). We make all of the top $K$ items have the same partworths while the bottom $n - K$ items have the same (and lower) partworths. The key challenge then is to obtain a tight upper bound on the KL divergence between any two hypothesis models. In summary, Theorem (5.1) implies the following *minimum* sample complexity for any algorithm for top-$K$ recovery:

$$\Omega\left( \frac{n \log n}{m \Delta_K^{(m)}} \cdot \left(1 + \frac{\tau_{K+1}^{(m)}}{\Delta_K^{(m)}}\right) \right).$$

Comparing with the bound in Theorem 4.4, one can see that the sample complexity of Borda Count is optimal in terms of both $m$, $n$ as well as the model dependent parameters $\Delta_K^{(m)}$ and $\tau_{K+1}^{(m)}$.

# 6 THE ROLE OF THE MENU SIZE $m$

The effect of the menu size on the performance of top-$K$ recovery algorithms is an aspect of both theoretical and practical importance. In real life applications, the menu size could range from 2 to hundreds of items. One may suspect that increasing the menu size means the data carries more information per data point, and thereby reduces the sample complexity for top-$K$ recovery. However, to the best of our knowledge, such a relationship has not been theoretically established in the literature on choice modeling, even for the very commonly used MNL model.

As seen in the matching lower and upper bound for the sample complexity of top-$K$ recovery, the menu size enters

in complex ways through the factors $\Delta_K^{(m)}$ and $\tau_{K+1}^{(m)}$. Both factors can vary in subtle ways with $m$, depending on the underlying choice model. Even for the class of MNL models which admit closed form choice probabilities, these factors don't seem to have a closed form expression as each of them is a sum of $\binom{n-1}{m-1}$ terms. To bypass the difficulty of exactly evaluating $\Delta_K^{(m)}$ and $\tau_{K+1}^{(m)}$, we characterize the asymptotic dependency of $\frac{1}{m\Delta_K^{(m)}}$ and $\frac{\tau_{K+1}^{(m)}}{\Delta_K^{(m)}}$ on $m$ *under the MNL class of models* and show that both of these factors monotonically decrease with $m$ but at a *decreasing rate*. This implies that while there is an advantage to using choice data of larger menu sizes, there is a diminishing return to increasing the menu size.

**Theorem 6.1.** *For any MNL model and a fixed $K$,*

$$\frac{1}{m\Delta_K^{(m)}} = \theta\left(\frac{1}{e^{U_K} - e^{U_{K+1}}} \cdot \left(1 + \frac{1}{m-1}\right)\right),$$

$$\frac{\tau_{K+1}^{(m)}}{\Delta_K^{(m)}} = \theta\left(\frac{e^{U_{K+1}}}{e^{U_K} - e^{U_{K+1}}} \cdot \left(1 + \frac{1}{m-1}\right)\right).$$

It can be seen in both $\frac{1}{m\Delta_K^{(m)}}$ and $\frac{\tau_{K+1}^{(m)}}{\Delta_K^{(m)}}$ that the term which depends on $m$, $1 + \frac{1}{m-1}$, montonically decreases with $m$ but at a diminishing rate. Combining the above theorem and the matching sample complexity bounds obtained earlier, one can see that the optimal sample complexity for top-$K$ recovery from choice data scales as $\theta(1 + \frac{1}{m})$.

Outside of the MNL family of models, we are not aware of any IID-RUM that admits a closed form expression for the choice probabilities. However, suppose that we know all of the partworths and the noise distribution, we can still approximate the choice probabilities via Monte Carlo sampling. Given these (approximated) choice probabilities, one can then evaluate $\frac{1}{m\Delta_K^{(m)}}$ and $\frac{\tau_{K+1}^{(m)}}{\Delta_K^{(m)}}$. As an example, Figure 1 shows how these quantities vary with $m$ under a randomly generated MNL and Probit model (IID-RUM with standard normal noise) [Train, 2009] with $n = 15$, $K = 3$. The partworths were independently generated from a zero-mean normal distribution which is also a commonly chosen prior in the literature [Parkes et al., 2012, Train, 2009]. The curves for $\frac{1}{m\Delta_K^{(m)}}$ and $\frac{\tau_{K+1}^{(m)}}{\Delta_K^{(m)}}$ decrease at a rate approximately similar to those of the MNL model as stated in Theorem 6.1. Ranking from choice data under MNL model assumptions remains an active area of research [Agarwal et al., 2018, 2020] and to the best of our knowledge, our work presents *the first asymptotic characterization* of the optimal sample complexity for top-$K$ recovery in terms of the menu size $m$ under this often used class of choice models.

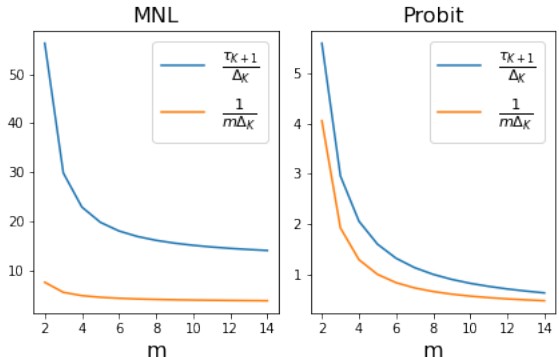

Figure 1: $\frac{1}{m\Delta_K^{(m)}}$ and $\frac{\tau_{K+1}^{(m)}}{\Delta_K^{(m)}}$ decrease with larger $m$ under a randomly generated MNL and Probit model. This suggests that there is an advantage, albeit with dimnishing return, to using larger menu sizes.

## 7 CONNECTIONS TO COMMONLY USED RANKING ALGORITHMS

In this section, we establish close connections among choice-based Borda count, the method of maximum likelihood estimate under MNL assumptions (MNL-MLE) [Train, 2009] and Spectral Ranking [Negahban et al., 2017, Maystre and Grossglauser, 2015, Agarwal et al., 2018] which will explain many experimental results we present in later sections.

Firstly, one can prove that choice-based Borda count and MNL-MLE are 'equivalent' top-$K$ recovery algorithms in the limit of infinite data. This connection is formalized as follows.

**Theorem 7.1.** *Consider the sampling model described in Section 4.1, for any $p > 0$, in the limit as $R \to \infty$, MNL-MLE and choice-based Borda count will produce the same top-$K$ estimate. Moreover, this holds even if the data does not come from the MNL model or any IID-RUM.*

A similar observation was made in Rajkumar and Agarwal [2014]: under the *pairwise comparison* setting the Borda count algorithm and MNL-MLE are both consistent for full ranking under a class of pairwise comparison models that is strictly more general than the BTL model [3]. Our results generalize the relation between the two algorithms to the choice setting and show that in fact the Borda count algorithm and MNL-MLE produce the same estimate in the limit of infinite data under any choice models. This connection between the two methods is reflected in our experiments where the performance of the choice-based Borda count algorithm is almost identical to that of MLE, when the sample size is large. While performing similarly to MNL-MLE, the choice-based Borda count algorithm is

---

[3]The BTL model is the instantiation of the MNL to the pairwise comparison setting

several orders of magnitude faster thanks to its simplicity. This suggests that if the statistician is mostly concerned with recovering a small number of top items, the choice-based Borda count algorithm should be seriously considered due to its speed, simplicity and guaranteed optimal sample complexity.

The above result also means that MNL-MLE is a consistent top-$K$ ranking algorithm under the broad class of IID-RUMs, since the choice-based Borda count algorithm is consistent in recovering the top $K$ items. This shows that MNL-MLE may be used for ranking applications even when the data does not satisfy the MNL assumption. Consistency of MLE under model misspecification is an underexplored question and we leave the careful characterization of the sample complexity of MNL-MLE when the data comes from a non-MNL distribution as a subject of future studies.

On the other hand, Spectral Ranking does not in general produce the same top-$K$ estimate as MNL-MLE/choice-based Borda count. However, when the underlying choice model falls within a broad class of IID-RUMs which include many commonly used choice models such as the MNL and Probit model, all three algorithms produce the same estimate given infinite data.

**Theorem 7.2.** *Consider the sampling model described in Section 4.1. Assume that the underlying choice model generating the data is in the class of IID-RUMs whose noise distribution has absolutely continuous density function with support on the real line. For any $p > 0$, in the limit as $R \to \infty$, then Spectral Ranking, MNL-MLE and choice-based Borda count produce the same top-$K$ estimate.*

*On the other hand, there exists a choice model where in the limit as $R \to \infty$, the Spectral Ranking algorithm produces a different top-$K$ estimate from MNL-MLE/Borda count.*

# 8 EXPERIMENTS

In this section, we present experiment results on both synthetic and real datasets. The main performance metric is top-$K$ accuracy. More specifically, we measure top-$K$ accuracy as the frequency at which the respective algorithm correctly identifies *all* of the true top $K$ items, *over 100 trials*.

## 8.1 SYNTHETIC DATA

We verify, via synthetic experiments, the efficacy of the choice-based Borda count algorithm and the effect of the menu size $m$ on its performance. Let there be $n = 50$ items in the universe. We experiment with 3 different noise distributions: standard Normal noise (Probit), standard Gumbel noise (MNL) and standard Exponential noise. We vary the menu size $m = 2, 4, 6, 8$ and $K = 1, 3, 5$. Figure 2 shows top-$K$ accuracy against the sample size. In all experiments,

choice-based Borda count successfully identifies the top $K$ items with high probability given sufficiently large sample size. Furthermore, using larger menu sizes improves the performance of Borda Count. However, it can be seen that there is a diminishing return in performance gains from using larger menu sizes, agreeing with our theoretical analysis in Section 5.

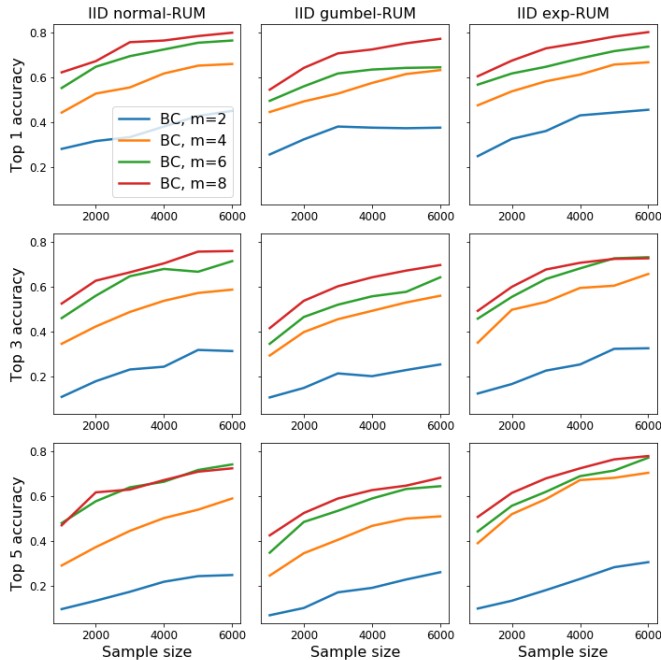

Figure 2: **Synthetic data:** Exact top $K$ accuracy of choice-based Borda count against sample size for different menu sizes 2 (blue), 4 (orange), 6 (green), 8 (red) with $K = 1, 3, 5$ and $n = 50$. Increasing the menu size improves performance but with a diminishing return.

## 8.2 REAL DATA

**Baseline algorithms:** We compare choice-based Borda count against Accelerated Spectral Ranking (ASR) [Agarwal et al., 2018] and Maximum Likelihood Estimate (MLE) under MNL assumptions [Train, 2009] in terms of top $K$ accuracy. We implement MLE using Scipy's L-BFGS optimizer [Virtanen et al., 2020].

**Data description:** We follow standard procedures commonly used in previous works such as Rajkumar and Agarwal [2014], Agarwal et al. [2020]. Operating on full ranking datasets, we can estimate the choice probabilities for any menu $S$, i.e., choice probability $\rho(i|S)$ is the proportion of rankings that ranks item $i$ highest among all them items in $S$. Given these probabilities, we can simulate the sampling model as described in Section 4.1. Our datasets include SUSHI [Kamishima, 2003], APA election dataset [Diaconis, 1989], 3 Irish election datsets and F1 race dataset included

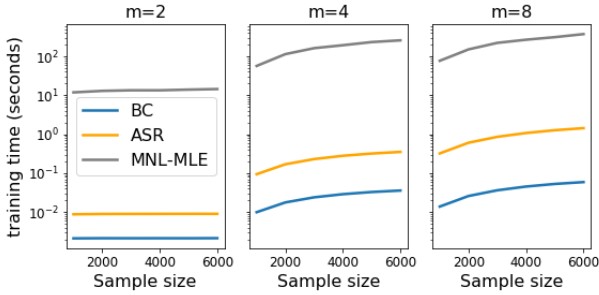

Figure 3: **F1 dataset** ($n = 22$)**:** Average training time (seconds) against sample size for $m = 2, 4, 8$. Choice-based Borda count (blue) is several orders of magnitude faster than its competitors.

in the library PrefLib [Mattei and Walsh, 2013]. Notably, the induced pairwise choice probabilities of these datasets all satisfy stochastic transitivity. Therefore, there exists a universal ordering of the items which we can use as a true global ranking over the items. Due to space constraint, we can only present a few representative experimental findings and leave additional results with detailed descriptions of data processing in the supplementary materials.

**Speed advantage:** Across all experiments, choice-based Borda count is several orders of magnitude faster than ASR and MLE. This difference is especially pronounced in datasets with more items such as the F1 dataset, as shown in Figure 3.

**Competitive accuracy:** Figure 4 show the performance of the algorithms under the Irish-Meath dataset and Figure 5 shows the results for the Irish-West dataset. Our theoretical analysis in Section 7 is reflected in our experimental findings: the performance of MNL-MLE and the Borda count algorithm are very similar given sufficiently large sample size. Spectral Ranking, on the other hand, may perform better or worse than MLE/Borda count depending on the dataset and the choice of $m$ and $K$. For many combinations of $m$ and $K$, we observe that the choice-based Borda count algorithm accurately recovers the top $K$ items and is highly competitive with MNL-MLE and Spectral Ranking. Notably, in most datasets, for smaller $K$ and large $m$, ithe Borda count algorithm has considerable advantages thanks to its accuracy and faster running time. In practice, this means choice-based Borda count is appropriate for applications where the statistician is interested in quickly determining a single (or a few) top candidate(s) from a large amount of data such as aggregating political surveys.

# 9 CONCLUSION

Ranking under Random Utility Models is a promising area of research with many practical applications. Our work shows how an efficient algorithm can perform very well

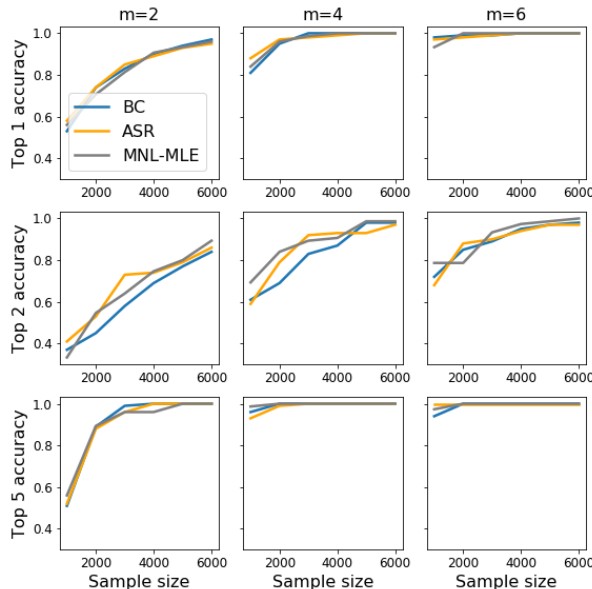

Figure 4: **Irish-Meath dataset** ($n = 14$)**:** Exact top-$K$ accuracy against sample size. choice-based Borda count (blue) is competitive with baseline algorithms. Using larger menu sizes generally improves the performance of the algorithms.

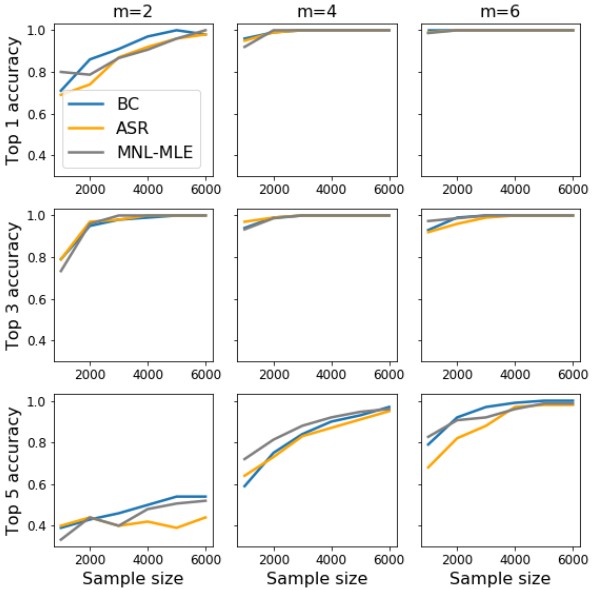

Figure 5: **Irish West dataset** ($n = 9$)**:** choice-based Borda count (blue) performs very similarly to MNL-MLE (gray) while the performance of ASR (orange) may diverge from MNL-MLE/choice-based Borda count.

under a broad family of RUMs. That being said, the class of IID-RUMs constitutes only a subset of models within the class of general RUMs. Beyond IID-RUMs, not much is known in terms of efficient inference and ranking algorithms. In the future, we hope to see more ranking methods developed for more expressive RUMs which have non-identical noise distributions or dependent noise distributions.

## Acknowledgements

The author thanks Shivani Agarwal for suggesting the idea of generalizing the Borda count algorithm to the choice setting, proofreading earlier versions of this paper and for helpful discussions. The author also thanks Prathamesh Patil and William Zhang for proofreading this paper; and the anonymous reviewers for their comments. This work is supported in part by the National Science Foundation (NSF) under grant number 1717290 (awarded to Shivani Agarwal). Any opinions expressed in this paper are those of the author and do not necessarily reflect the views of the National Science Foundation.

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
