# OpenReview forum: "Efficient and Accurate Top-$K$ Recovery from Choice Data"
_auai.org/UAI/2022/Conference — UAI 2022 Poster_

### Official Review · Reviewer_gRxA · 2022-04-09

**Q2(1) Originality/Novelty:** 3
**Q2(2) Significance/Impact:** 2
**Q2(3) Correctness/Technical Quality:** 3
**Q2(6) Clarity Of Writing:** 3
**Q6 Overall Score:** 7
**Q8 Confidence In Your Score:** 3

**Q1 Summary And Contributions:**

This paper proposes the choice-based Borda count algorithm to efficiently solve the top-K recovery problem. The authors extends the Borda count idea to the discrete choice setting, and prove that the algorithm can reach the optimal sample complexity for MNL models. The authors also show that the Borda count algorithm produces the same top-K prediction as the MLE method for MNL models but with less running time.

**Q2 Assessment Of The Paper:**

More detailed information regarding each of these aspects is given below:

**Q2(4) Quality Of Experiments (Optional):**

2: Fair: The experimental evaluation is weak: important baselines are missing, or the results do not adequately support the main claims.

**Q2(5) Reproducibility:**

3: Good: Key resources (e.g., proofs, code, data) are available and key details (e.g., proofs, experimental setup) are sufficiently well-described for competent researchers to confidently reproduce the main results.

**Q3 Main Strengths:**

The conclusion is significant, as the authors proposed an algorithm which is both easy to implement and with the optimal sample complexity for some models (e.g. MNL). The authors also showed that, both in experiments and in the proof, that the algorithm will provide the same top-K estimate as the common approach (MLE) for MNL models under assumptions, but the proposed algorithm has a much faster speed.

The proof sketch is also clearly written. The authors showed the conclusion mainly by a concentration bound with some further computations on the parameters. The authors also provided an analysis on the influence of the menu size $m$, which proposed useful insights for practical usages.

**Q4 Main Weakness:**

One major point to improve is on the experiment settings. In the experiments, the authors only used very small values for the universe size $n$, the menu size $m$ and $K$. In my opinion, Algorithm 1 can be applied to relatively large scale applications. Only showing results for $n\le22$ and $m\le10$ might be insufficient to demonstrate the empirical performance for the algorithm since the authors also mentioned that top-K retrieval algorithms are useful in applications like recommendation systems.

Another point is on the sample complexity lower bound. The authors studied the sample complexity of Algorithm 1 on more general cases but only work on the sample complexity lower bound specifically on MNL models. What sample complexity lower bound will we have if we have a noise other than the Gumbel noise? I understand it might be hard to get exactly the same bound but a weaker bound is also appreciated if it is possible for the more general cases.

**Q5 Detailed Comments To The Authors:**

Hope the authors can perform some larger scale experiments and some lower bound analysis as I mentioned in Q4. Besides, it will be better if the authors can add more introduction for baseline and existing methods. For example, the authors used Spectral Ranking in the experiments, but did not introduce it in the main paper.

**Q7 Justification For Your Score:**

The proposed algorithm is both efficient and optimal on the sample complexity for some cases. Besides, the authors also have additional analysis (i.e. the analysis on $m$) that provides useful insights for practical applications.

**Q9 Complying With Reviewing Instructions:**

1: Yes.

---

### Official Review · Reviewer_iWoT · 2022-04-12

**Q2(1) Originality/Novelty:** 2
**Q2(2) Significance/Impact:** 2
**Q2(3) Correctness/Technical Quality:** 2
**Q2(6) Clarity Of Writing:** 3
**Q6 Overall Score:** 6
**Q8 Confidence In Your Score:** 3

**Q1 Summary And Contributions:**

This paper proposes a choice-based Borda count algorithm that aims at providing a faster (with optimal sample complexity) yet accurate ranking algorithm fpr top K recovery. The proposed algorithm is for top-k recovery, this field has been investigated extensively and is widely used in several fields such as item recommendation and top candidates filtering.

**Q2 Assessment Of The Paper:**

More detailed information regarding each of these aspects is given below:

**Q2(4) Quality Of Experiments (Optional):**

3: Good: The experimental evaluation is adequate, and the results convincingly support the main claims.

**Q2(5) Reproducibility:**

2: Fair: Key resources (e.g., proofs, code, data) are unavailable but key details (e.g., proof sketches, experimental setup) are sufficiently well-described for an expert to confidently reproduce the main results.

**Q3 Main Strengths:**

1.  The paper is well-written and I found it to be easy to follow. The claims and the propositions in this paper are discussed in both Main body and appendix section which are reasonable, additional experiments with various settings support the effectiveness of proposed method.
2. The time complexity of proposed BC algorithm has been lowered compared with the other methods while preserving the accuracy in top-k recovery accuracy.

**Q4 Main Weakness:**

1. Regarding to the training time, which is one of the largest contribution that this paper claimed, however, there is only Figure 3 demonstrating the improving speed in one dataset, which I found it to be limiting. Adding a few more experiments showing the training time comparison will make the time complexity improvement more convinincing.
2. I found that there are a few confusing notations: in Figure 1, m denotes the size of the displayed menu, the values in the x axis should be integers.


**Q5 Detailed Comments To The Authors:**

As discussed in the above weakness section.

**Q7 Justification For Your Score:**

I only read the main body of the paper and skimmed through the appendix.

**Q9 Complying With Reviewing Instructions:**

1: Yes.

---

### Official Review · Reviewer_m2qa · 2022-04-12

**Q2(1) Originality/Novelty:** 3
**Q2(2) Significance/Impact:** 3
**Q2(3) Correctness/Technical Quality:** 3
**Q2(6) Clarity Of Writing:** 2
**Q6 Overall Score:** 7
**Q8 Confidence In Your Score:** 3

**Q1 Summary And Contributions:**

The authors propose a choice-based Borda count algorithm for fast and accurate ranking for top-K recovery.
The main contribution of the paper is showing a lower sample complexity in the context of random utility models. For this, the authors define a quantity that they call “Borda consistency” and that represents the expected probability that an item i is chosen from a menu S of m elements.
Further, their approach allows to generalise the framework to menus of variable size.

**Q2 Assessment Of The Paper:**

More detailed information regarding each of these aspects is given below:

**Q2(4) Quality Of Experiments (Optional):**

3: Good: The experimental evaluation is adequate, and the results convincingly support the main claims.

**Q2(5) Reproducibility:**

3: Good: Key resources (e.g., proofs, code, data) are available and key details (e.g., proofs, experimental setup) are sufficiently well-described for competent researchers to confidently reproduce the main results.

**Q3 Main Strengths:**

The paper is sound and reasonably impactful.
Interesting Model


**Q4 Main Weakness:**

A point that is left quite not tackled is any possible dependency of the method complexity on the size K of elements selected as the most highly ranked ones. K appears as an index in the calculations of the work and not as a quantity, however there is never really a point in the paper where a reflection on this is made. While some experimental simulations seem to account for different top K accuracies, there is no proper description of this setting. This creates some confusion over whether the top K accuracy could be indeed a parameter that influences the complexity of the problem or not and a clarification on this would be needed.


**Q5 Detailed Comments To The Authors:**

The paper is well presented, the notation is easy to follow and overall the model and the objectives are described clearly.
The contribution of the paper is quite relevant and many details have been thought out and accounted for both in the theoretical and the experimental part. I appreciated the discourse over the difficulty of establishing a clear dependency of the quantities on the value m.


**Q7 Justification For Your Score:**

The paper is sound and reasonably impactful.


**Q9 Complying With Reviewing Instructions:**

1: Yes.

---

### Official Review · Reviewer_6YE7 · 2022-04-13

**Q2(1) Originality/Novelty:** 2
**Q2(2) Significance/Impact:** 2
**Q2(3) Correctness/Technical Quality:** 3
**Q2(6) Clarity Of Writing:** 3
**Q6 Overall Score:** 4
**Q8 Confidence In Your Score:** 3

**Q1 Summary And Contributions:**

This paper proposes an approach to tackle the top-K ranking problem. This could be very helpful in domains like recommender systems, information retrieval, ... The paper mainly focuses on the Borda count algorithm and its connection with other approaches. The authors used many proofs to demonstrate the importance and feasibility of the proposal. The article is well written and easy to read.

**Q2 Assessment Of The Paper:**

More detailed information regarding each of these aspects is given below:

**Q2(4) Quality Of Experiments (Optional):**

2: Fair: The experimental evaluation is weak: important baselines are missing, or the results do not adequately support the main claims.

**Q2(5) Reproducibility:**

2: Fair: Key resources (e.g., proofs, code, data) are unavailable but key details (e.g., proof sketches, experimental setup) are sufficiently well-described for an expert to confidently reproduce the main results.

**Q3 Main Strengths:**

- Mathematical proofs of every aspects of the proposal, this makes the reader more confident to the proposal
- Paper is well organized

**Q4 Main Weakness:**

- The originality of the proposal is marginal, as Borda count is a very simple approach. Even its generalization here doesn't consider dependancies among items.
- The contribution have moderate impact in real-world application.
- Experimentations are limited to small size datasets, even if the approach is devoted to large choice datasets as claim by the authors in the introduction.


**Q5 Detailed Comments To The Authors:**

This paper proposes an approach to tackle the top-k elements in the ranking problem. This could be very helpful in domains like recommender systems, information retrieval, ... The document mainly focuses on the Borda count algorithm and its connection with other approaches. The authors used many proofs to support the importance and feasibility of the proposal. Experimentations on small choice datasets (synthetic, real) are also described . This very long article (29 pages of supplemental materials) is well written.

My main concern here is the choice of the simple Borda count algorithm for top-K ranking problem, where many dependences may occur among items. Just considering the frequency (support count) is too limited.

Moreover experimentations are limited to small size datasets, even if the approach is devoted to large choice datasets as claim by the authors in the introduction.

In addition, the paper does not clearly tackled the uncertainty aspect in this task.

Minor :
- I wonder that K is not an input of the algorithm 1.
- Several references are incomplete.

**Q7 Justification For Your Score:**

The contribution is marginal.

**Q9 Complying With Reviewing Instructions:**

1: Yes.

---

### Decision · Program_Chairs · 2022-05-15

**Decision:**

Accept (Poster)

**Comment:**

Meta Review: This paper studies the problem of top-K recovery from a fixed set of queries. In each query, the user is presented m items and they select the item with the highest noisy utility. The authors propose a simple solution to this problem, take top-K most frequently chosen items, and prove that it is near optimal. The proposed solution is simple and valid only under strong assumptions, the utility of the query is additive in its items and the item utility noise is independent. Nevertheless, the problem is studied in depth, including a lower bound and comparison to model-based approaches to solving the problem. All reviewers liked the paper and I support acceptance.